
# Testing the altitude attribution and vertical resolution of AirCore measurements with a new spiking method

Thomas Wagenhäuser[1], Andreas Engel[1], Robert Sitals[1]

[1]Institute for Atmospheric and Environmental Sciences, University of Frankfurt, Frankfurt, 60438, Germany

*Correspondence to*: Thomas Wagenhäuser (wagenhaeuser@iau.uni-frankfurt.de)

**Abstract.** AirCores have been increasingly used to capture vertical profiles of trace gases reaching from the ground up to about 30 km, in order to validate remote sensing instruments and to investigate transport processes in the stratosphere. When deployed to a weather balloon, accurately attributing the trace gas measurements to the sampling altitudes is non-trivial especially in the stratosphere. In this paper we present the CO-spiking experiment, which can be deployed to any AirCore on

any platform in order to evaluate different computational altitude attribution processes and to experimentally derive the vertical resolution of the profile by injecting small volumes of signal gas at predefined GPS-altitudes during sampling. We performed two CO-spiking flights with an AirCore from the Goethe-University of Frankfurt (GUF) deployed to a weather balloon in Traînou, France in June 2019. The altitude retrieval based on an instantaneous pressure equilibrium assumption slightly overestimates the sampling altitudes, especially at the top of the profiles. For these two flights our altitude attribution is accurate

within 250 m below 20 km. Above 20 km the bias becomes larger and reaches up to 1.2 km at 27 km altitude. Differences in descent velocities are uncovered to have a major impact on the altitude attribution bias. We identified the time lag between the theoretically attributed altitude and the actual CO-spike release altitude to be a possible empirical correction parameter for our AirCore altitude retrieval across different flights. Regarding the corrected profiles, the altitude attribution is accurate within 120 m throughout the profile. Further investigations are needed in order to test for the scope of validity of this correction

parameter regarding different ambient conditions and maximum flight altitudes. We derive the vertical resolution from the CO-spikes of both flights and compare it to the modelled vertical resolution. The modelled vertical resolution is better than the experimentally derived resolution throughout the profile, albeit agrees within 220 m. All our findings derived from the two CO-spiking flights are strictly bound to the GUF AirCore dimensions. The newly introduced CO-spiking experiment can be used to test different combinations of AirCore configurations and platforms in future studies.

## 1 Introduction

The AirCore is a cost-effective atmospheric sampling technique originally developed by Pieter Tans (2009) and introduced by Karion et al. (2010) to capture vertical profiles of trace gases. In principle, it consists of a coiled stainless steel tube, that is sealed at one end and open at the other. During ascent, e.g. on a weather balloon, it empties due to the decreasing pressure with height, whereas during descent the surrounding air flows into the AirCore. After recovery the sample is analyzed for trace gas



mole fractions with a continuous gas analyzer and the resulting measurements are attributed to the sampling altitudes. AirCores have been increasingly deployed to small weather balloons to capture continuous $CO_2$ and $CH_4$ profiles from the surface up to about 30 km at various locations around the world. Recently, AirCore measurements have been used to validate ground-based spectrometric data of the Total Carbon Column Observing Network (TCCON) (Sha et al., 2020; Tu et al., 2020), which is widely used to validate satellite data. Vertical information, that has been derived from ground-based remote sensing, has been

compared with AirCore profiles (Karppinen et al., 2020; Zhou et al., 2019). Tadić and Biraud (2018) used AirCore data to evaluate their approach to estimate total column mole fractions of $CO_2$ and $CH_4$ using partial column data from aircraft flights. Further developments based on the AirCore sampling technique allow for new areas of application. E.g. Andersen et al. (2018) developed an active AirCore sampling system and deployed it to a light-weight unmanned aerial vehicle for tropospheric sampling at locations that are difficult to access. Instead of passively sampling ambient air due to the increase in ambient

pressure during descent, they used a pump to actively pull ambient air through their AirCore. Karion et al. (2010) proposed to deploy AirCores to maneuverable gliders, which would facilitate probing specific air masses and recovering the AirCore. AirCore subsampling techniques have been developed that allow for analysis of isotopes (Mrozek et al., 2016; Paul et al., 2016) and halogenated trace gases with abundances well below 1 part per billion (Laube et al., 2020). Stratospheric trace gas measurements play an important role to investigate dynamical changes in the stratosphere (Moore et al., 2014). Engel et al.

(2017) derived the mean age of air at high altitudes from AirCore measurements in order to update their investigation of long-term changes in the overall overturning circulation of the stratosphere (Brewer–Dobson circulation) based on atmospheric observations presented in Engel et al. (2009). Their AirCores were deployed to a large stratospheric balloon launched by CNES in 2015 and to small weather balloons flown in 2016.

The wide range of platforms and fields of application concerning AirCore sampling all have one in common: a continuous

sample of atmospheric air is collected that needs to be attributed to positional data. Regarding vertical profiles from passive AirCores, an altitude attribution approach has been suggested (P. Tans, NOAA, private communication, 2020), that is based on modelling the pressure drop across the AirCore during sampling and the flow of air into the AirCore. However, until now a common approximation is to assume an instantaneous pressure equilibrium between the AirCore and ambient air and to use the ideal gas law to calculate the amount of sample for each time step during descent (e.g. Engel et al., 2017; Karion et al.,

2010; Membrive et al., 2017). In addition, the start and end points of the AirCore in the continuous trace gas measurement time series need to be determined accurately, which relied on subjective judging until now (Engel et al., 2017; Membrive et al., 2017). The reliability of assuming an instantaneous pressure equilibrium during sampling for the altitude attribution depends on multiple factors (e.g. AirCore geometry, usage of a dryer, ascent and descent velocities, magnitude of pressure change with altitude). Due to the weak vertical pressure gradient at high altitudes, especially attributing the stratospheric part

of balloon-based AirCore observations to the correct altitude is a challenging task and it can be considered even more challenging when descent velocities are high. The latter is the case for descents that are decelerated solely by parachutes, which is the most common way for AirCores flown from weather balloons. To our knowledge, the altitude attribution processes could only be validated by comparing AirCore profiles to in situ aircraft measurements and sampling flasks up to approximately



350 hPa (Karion et al., 2010) – corresponding to below 10 km – or comparison of different AirCores on a slowly descending large stratospheric balloon (Engel et al., 2017; Membrive et al., 2017) or comparison with a lightweight stratospheric air sampler (Hooghiem et al., 2018), without quantifying any altitude attribution bias until now.

In this paper, we present a CO-spiking system, a newly developed technique that can be used in situ to evaluate any combination of AirCore, platform and altitude retrieval procedure. In principle, this technique could also be used to evaluate the positional retrieval of active AirCores. Here, we focus on a passive AirCore that has been deployed to a weather balloon and on the commonly applied retrieval procedure that is based on assuming an instantaneous pressure equilibrium. In Section 2, we describe the AirCore and analytical set-up, together with the retrieval procedure that we use and the technical CO-spiking setup. Two measurement flights, that were conducted in Traînou, France in June 2019 are presented in Sect. 3 together with the CO-spiking experiment results regarding altitude attribution, a possible correction parameter and the vertical resolution of the profiles at different altitudes. We summarize the findings of this paper and give conclusions in Sect. 4.

## 2. Experimental

### 2.1 AirCore Goethe-University Frankfurt general approach

The Goethe-University Frankfurt (GUF) AirCore is designed to be light weight and is thus allowed for use under small balloons at mid-latitudes in Europe. Its geometry promotes high sampling volume and reduces mixing due to diffusion during storage especially in the high-altitude sampling region, where tubing with thinner inner diameter is used. The experimental set-up, operation and data evaluation of the GUF AirCore have been published in detail in Engel et al. (2017).

Three thin-walled stainless steel tubes with different outer diameters (O.D. 2 mm, 4 mm, 8 mm), coated with silconert2000® and soldered together result in the 100 m long and coiled GUF AirCore. This design allows to rapidly collect air in the large-diameter tubing, which is then gradually pushed into the smaller-diameter tubing, where mixing by molecular diffusion is less effective. A dryer filled with $Mg(ClO_4)_2$ is connected to the inlet. The onboard electronic system that we used from 2019 on is based on the Arduino MEGA 2560 micro controller. It comprises up to 8 temperature sensors, a pressure sensor, a GPS-antenna, an SD-card holder for data logging and controls the closing valve.

Before launch, the AirCore is flushed with fill gas (FG) and sealed at one end. During ascent it empties due to the decreasing ambient pressure with height. A small amount of FG remains in the AirCore. During descent the AirCore fills with ambient air due to the increase in ambient pressure. Upon landing, the inlet is closed automatically. After retrieval the AirCore is brought back to the laboratory. The sample is pushed out of the AirCore with a push gas (PG) and analyzed with a Picarro G2401 CRDS continuous gas analyzer for $H_2O$, CO, $CH_4$ and $CO_2$ mole fractions. Figure 1 shows the analytical set-up for the measurement process. Since June 17, 2019, our Picarro analyzer operated in the inlet valve control mode at a constant rate at ~30 sccm for AirCore measurements. This is similar to previously published operating methods (e.g. Andersen et al., 2018; Membrive et al., 2017). The mass flow controller in the original measurement setup described in Engel et al. (2017) was replaced by a needle valve (NV) acting as a flow resistance close to the inlet of the Picarro analyzer. This set-up has the





advantage that the mass flow controller, which provides an additional source of mixing before the analysis cell, can be removed.

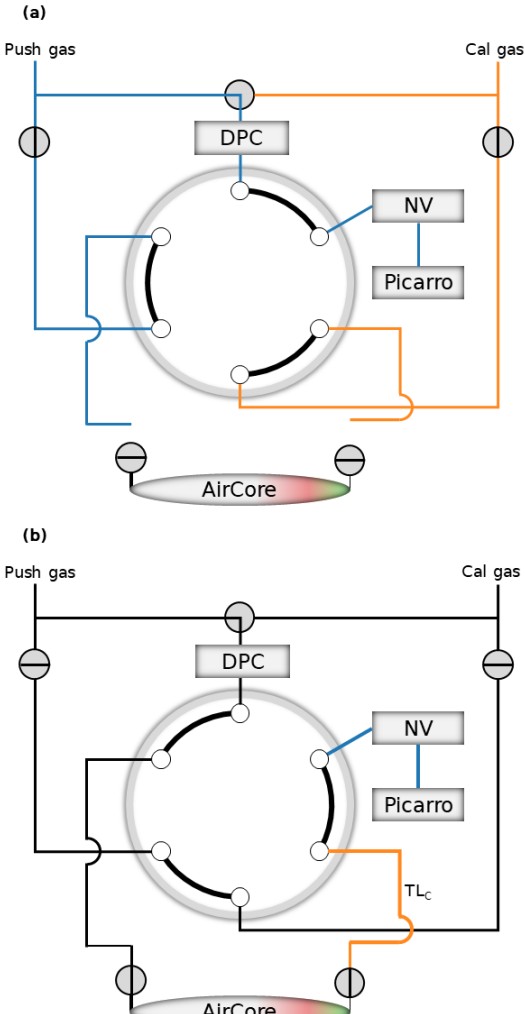

**Figure 1:** Analytical set-up for AirCore measurements. Pressure is controlled by the digital pressure controller (DPC). Compared to the previously published set-up by Engel et al. (2017) the mass flow controller has been replaced by a needle valve (NV). The Picarro operates in inlet control mode. In the bypass/flushing position **(a)** push gas (PG) is measured bypassing the AirCore while the transfer lines are being flushed with PG resp. a calibration standard (Cal gas). Tubing that contains PG is indicated in blue, Cal gas in orange. In the AirCore measurement position **(b)** the PG is passed through the AirCore and pushes the air to the Picarro. Directly after switching to **(b)** a small amount of Cal gas is measured that has been enclosed by the transfer line $TL_C$. For clearness, in **(b)** only tubing involved at the start of the AirCore measurement is coloured (adapted from Engel et al., 2017).

When deployed to a weather balloon, a retrieval procedure is required, which attributes the measured trace gas mixing ratios to the sampling altitudes in order to retrieve a vertical profile. Our retrieval procedure is a three-stage process, which is re-ordered and refined compared to the four-stage process described in Engel et al. (2017). The overall concept of the retrieval





remains the same. (i) The sampling of air during the balloon descent is calculated based on the ideal gas law (see Engel et al.
(2017) Sect. 2.4.2 for details). (ii) The start and end times of the AirCore measurement in the analyzer time series are
determined. (iii) The sampling and the analysis can be matched based on the molar amount following Engel et al. (2017). Steps
(i) and (iii) are still performed according to Engel et al. (2017). In Sect. 2.2 we present a new approach to determine the start
point of the AirCore in the measured trace gas time series. This new approach has the advantage of providing an objective start
point without the need for subjective judging.

## 2.2 Start point determination


Membrive et al. (2017) stated that for their slowly descending high resolution AirCore the dominating uncertainty source in
the stratosphere is related to the selection of the AirCore starting point in the analysis data. They link this to the low amount
of stratospheric sample compared to the tropospheric sample. For AirCores with less stratospheric sample the effect can be
considered to be larger. Until now, the choice of the starting point relied on subjective judging (Engel et al., 2017; Membrive
et al., 2017). In order to systematically evaluate the altitude attribution procedure with the CO-spiking experiment presented
in this paper, as many as possible subjective parameters need to be eliminated. We therefore decided to refine the process of
selecting the start time of the AirCore and introduce a new approach to identify an accurate starting point without the need for
subjective judging.

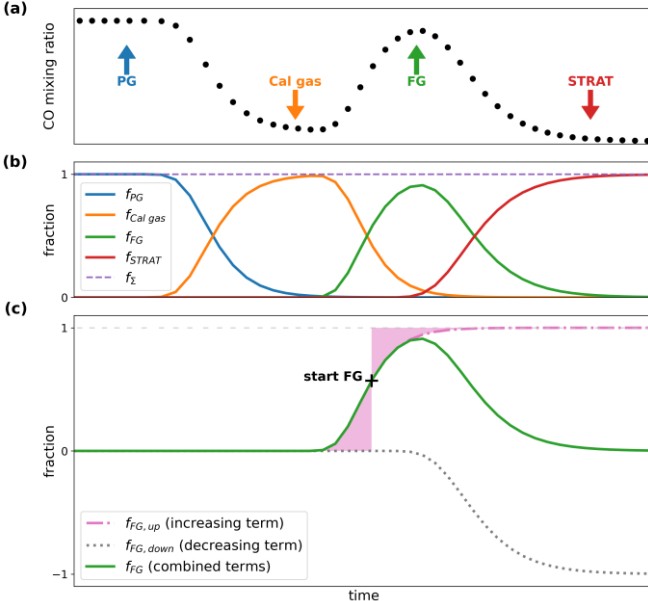

**Figure 2:** Idealized time series at the start of a GUF AirCore measurement with a Picarro CRDS. **(a)** CO mixing ratio time series. **(b)** Gas
fraction time series of push gas (PG), calibration gas (Cal gas), fill gas (FG) and stratospheric sample (STRAT) corresponding to **(a)**. $f_\Sigma$ is
the sum of all fractions and always 1. **(c)** Example time series of increasing and decreasing term for the FG fraction. At the starting point
"start FG", the two areas of the increasing term indicated in pink are of equal size.



For a regular GUF set-up AirCore flight analysis we first measure PG (high CO). We then switch the two position valve (see

Figure 1b) like described in Engel et al. (2017) so that secondly the calibration gas (Cal gas) within the transfer line $TL_C$ between AirCore and analyzer is measured (low CO). The Cal gas is used to distinguish between PG and FG. Thirdly, it is followed by the remaining FG in the AirCore (high CO), which is fourthly followed by the stratospheric sample (STRAT, low CO). The resulting idealized CO mixing ratio time series is shown in Figure 2a. In the past, a Gaussian distribution was fitted to the FG peak in the CO measurements. The half maxima of the fit were then considered the start of the AirCore, respectively

the start of the STRAT, as described in Engel et al. (2017) Sect. 2.4.1 and 2.4.3. This FG peak however is partly mixed with PG, Cal gas and STRAT. In the new approach, we reconstruct the gas fraction time series for each of these gases, in order to separate the amount of PG, Cal gas, FG and STRAT based on the measured CO-signal. This is possible due to the fact, that a sequence of gases with known CO mole fractions is measured with a constant molar flow in the considered time interval. The fraction of each gas changes during time. Figure 2b shows the idealized gas fraction time series for these four gases. We

describe the fraction $f_i$ of each gas $i$ by a combination of two terms, one being an increasing ($f_{i,up}$) the other being a decreasing term ($f_{i,down}$, see Figure 2c for an example corresponding to FG). This is also expressed in Eq. (1):

$$f_i(t) = f_{i,up}(t) + f_{i,down}(t). \tag{1}$$

$f_{i,up}$ ranges from 0 to 1, whereas $f_{i,down}$ ranges from 0 to -1. For mass constancy, the decreasing term of one gas equals the increasing term of the subsequent gas multiplied by -1:

$$f_{i,down}(t) = -f_{i+1,up}(t) \tag{2}$$

The increasing term of the first gas equals 1, i.e. the PG measurements at the beginning of the AirCore measurement are considered to be stable. Likewise, the decreasing term of the last gas equals 0. This way, the sum of the gas fractions of all the gases always equals 1:

$$f_\Sigma(t) = \sum_{i=1}^{n} f_i(t) = 1, \tag{3}$$

where n is the number of considered gases. For a regular GUF set-up AirCore flight n equals 4 (i.e. PG, Cal gas, FG and STRAT). The gas fraction time series of all of the gases multiplied by their respective CO mixing ratio $\chi_{i,CO}$ yields the actual measured CO mixing ratio time series of the Picarro $\chi_{CO}(t)$:

$$\chi_{CO}(t) = \sum_{i=1}^{n} f_i(t) \cdot \chi_{i,CO} . \tag{4}$$

$\chi_{i,CO}$ of STRAT is approximated for each flight by taking the mean of a measurement section of the AirCore, that is

subjectively considered to be as unaffected as possible from mixing with FG and tropospheric air.

Gkinis et al. (2010) and Stowasser et al. (2014) used the cumulative distribution function (CDF) of a lognormal distribution to fit a step-wise change in mixing ratios that is smoothed only by mixing in the analyzer cell of a Picarro CRDS. In our case we found, that the transition from one gas fraction to the next can be well described by a CDF of a Gumbel distribution:

$$-f_{i,down}(t) = f_{i+1,up}(t) \approx e^{-e^{-(t-\mu_i)/\beta_i}}, \tag{5}$$

where μ is the mode of the respective Gumbel distribution (i.e. the inflection point of the CDF) and β is a measure for the standard deviation. For simplicity of the fitting process we decided to use the CDF of the Gumbel distribution instead of the



CDF of the lognormal distribution. By altering the parameters $\mu_i$ and $\beta_i$ simultaneously for each gas taking into account Eq. (5), a CO mixing ratio time series is calculated following Eq. (1) and then Eq. (4) and fitted to the CO mixing ratio time series actually measured by the Picarro analyzer. The start of the remaining FG $t_{FG}$ in the measurements is the point in time, when the integral over the respective increasing term equals the integral over the remaining associated decreasing term:

$$\int_{-\infty}^{t_{FG}} f_{FG,up}(t)dt = \int_{t_{FG}}^{\infty} \big(f_{Cal\ gas,down}(t) + 1\big)dt. \tag{6}$$

In other words, $t_{FG}$ is the point in time, when the amount of already passed FG (and possibly stratospheric sample) through the measurement cell equals the amount of remaining Cal gas (and possibly PG) in the cell (see also Figure 2c, "start FG"). Accordingly, the start of stratospheric sample $t_{AC}$ is the point in time, when

$$\int_{-\infty}^{t_{AC}} f_{STRAT,up}(t)dt = \int_{t_{AC}}^{\infty} \big(f_{FG,down}(t) + 1\big)dt. \tag{7}$$

The end time of the AirCore measurement in the analyzer time series $t_{stop}$ is determined manually from half the transition between tropospheric sample and PG in the CO analysis data. The gas fraction time series of the FG $f_{fillgas}(t)$ can be integrated over time in order to apply a sampling correction to the altitude retrieval procedure similar to Engel et al. (2017), Sect. 2.4.1. Albeit, we found this correction to only have a small impact on the resulting profiles and decided to exclude it from our retrievals. Instead, we use $t_{FG}$ as the starting point of the whole AirCore.

## 2.3 CO-spiking system set-up

The CO-spiking system is an experimental set-up, which can be temporarily added to any AirCore for a flight in order to evaluate the final altitude attribution. Small amounts of signal gas are pulsed in the inlet of the AirCore during descent at predefined GPS altitudes. When assigning the trace gas measurements to the sampling altitude, the CO-spike signals are assigned to a modelled altitude as well. The quality of the altitude retrieval can be evaluated by comparing the retrieved CO-spike altitudes to the release altitudes.

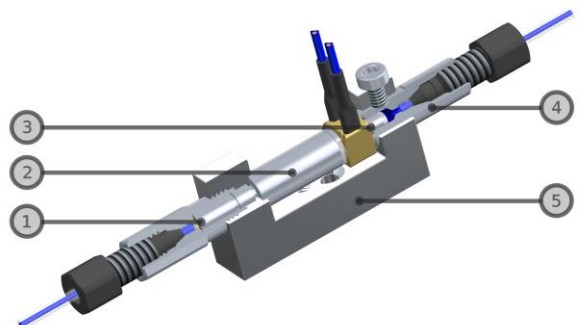

**Figure 3:** Fastening type for the SMLD 300G micro valve. (1) micro valve, (2) valve coil, (3) O-ring (material: viton), (4) inlet adapter, (5) valve holder. Fritz Gyger AG 2020.

The CO-spiking set-up consists of a signal gas reservoir, a micro valve and a connector which directly connects the micro valve to the open end of the AirCore in front of the sample drier. The adaptor is designed to have a negligible flow resistance


for sampled air, while inducing only a minimal dead volume to the sampling system. We used the micro valve SMLD 300G by Gyger, which is light-weight and suited to dose signal gas volumes of around ¼ ml$_n$ on the time scale of 20–100 milliseconds, thereby influencing the sampling process during descent as little as possible. Figure 3 illustrates the fastening
type for the micro valve.

We utilized a modified nylon compression ring and an additional O-ring (Figure 3, (3)) to achieve leak-tightness at low-temperature. In addition, the micro valve is heated during flight in order to remain leak-tight and maintain its functionality. The signal gas reservoir consists of a 2 m tubing (total volume approximately 50 ml) and a particle filter. The particle filter protects the micro valve from particles that might have entered the signal gas reservoir. The signal gas reservoir is coiled and
packed together with the AirCore in the Styrofoam box and directly connected to the micro valve. It can be pressurized via a valve at the other end of the tube and flushed by activating the micro valve. For flight preparation, the signal gas reservoir is pressurized from a signal gas canister with approximately 4 bar, flushed by activating the micro valve and then pressurized again. The signal gas has a very high mixing ratio of CO (in our case approx. 90 ppm) compared to typical atmospheric mixing ratios, so that measurable and discernible spikes can be generated with very small volumes of signal gas. During flight the
micro valve is controlled by the custom made AirCore onboard electronic system to release signal gas spikes at predefined GPS altitudes. After the retrieval, the AirCore sample is analyzed for trace gas mole fractions and attributed to the meteorological and altitude data like a regular AirCore sample e.g. by following Sect. 2.1. When determining the starting point of the sample during analysis of the AirCore CO-spiking flight data following Sect. 2.2, the first CO-spike is included in the gas fraction reconstruction process, as it can still overlap with the descending FG. Hence n in Eq. (3) equals 6 (i.e. PG, Cal
gas, FG, STRAT, signal gas, STRAT2).

## 3. Results

### 3.1 Measurement flights

We conducted two CO-spiking flights with our AirCore GUF003 during the AirCore campaign in Traînou in June 2019. This was one of two intensive AirCore campaigns in context of the EU-funded Readiness of Integrated carbon observation system
(ICOS) for Necessities of integrated Global Observations (RINGO) project.

The AirCore (GUF003) was prepared following Engel et al. (2017). In addition, the CO-spiking system was set up following Sect. 2.2. The FG which we also used as PG, contains high CO (approx. 1.4 ppm) relative to clean atmospheric air in order to be well distinguishable from both tropospheric and stratospheric air. The Cal gas contains approx. 0.15 ppm CO and is used to distinguish between PG and FG at the beginning of the AirCore analysis. The gas mixture that was utilized as signal gas
contains approximately 90 ppm of CO, which is almost two orders of magnitude higher than the CO in the FG. The micro controller was programmed to open the micro valve at predefined GPS altitudes for a certain amount of milliseconds. Table 1 lists the release altitudes and micro valve open times for each CO-spike.



**Table 1:** Release altitudes [km] and micro valve open times [ms] for the different spikes for the two flights on June 17 and June 18, 2019 in Traînou.

| Spike number | 1 (27 km) | 2 (22 km) | 3 (18 km) | 4 (15 km) | 5 (12 km) | 6 (9 km) | 7 (6 km) | 8 (3 km) |
|---|---|---|---|---|---|---|---|---|
| Micro valve open times / ms | 30 | 20 | 20 | 30 | 30 | 50 | 100 | 50 |
| Release altitude (June 17) / km | 27 | 22 | 17.99 | 14.99 | 11.99 | 9 | 6 | 3 |
| Release altitude (June 18) / km | 27 | 21.98 | 17.99 | 15 | 12 | 9 | 6 | 3 |


The first flight was on June 17, launching time 08:25 UTC. The payload was 3.5 kg and comprised the AirCore GUF003, a M10 radiosonde and a large parachute. The balloon burst at 09:50 UTC at 33.3 km and the payload landed 51 minutes later. The AirCore was brought back to the laboratory and started to be analyzed 2 h after landing. The second flight was on June 18, launching time 07:59 UTC. The payload was similar to the first flight, but instead of a large parachute, two smaller ones

were used. The balloon burst at 09:27 UTC at 33.2 km and the payload landed 41 minutes later. The analysis started 2 h after landing. The vertical pressure profiles were calculated from in situ GPS and temperature measurements, following Dirksen et al. (2014). The descent phase of the second flight was 10 minutes shorter than the one of the first flight, although both reached a similar altitude and the weather conditions were similar. The flight on June 18 thus had a higher descent rate than the flight on June 18. The descent velocity was calculated from the smoothed GPS altitude profile. Figure 4 shows the descent velocity

for the two flights versus altitude.

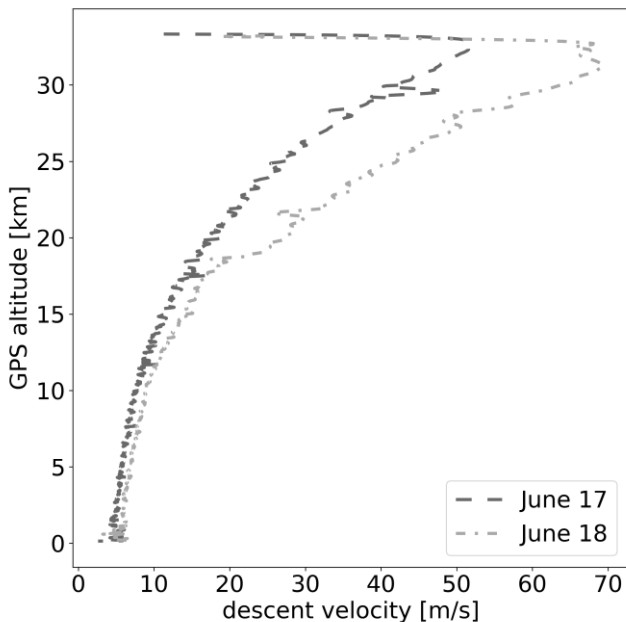

**Figure 4:** Descent velocities for the CO-spiking flights on June 17 (dark, dashed) and June 18 (light, dashdotted), Traînou 2019. The data is smoothed with a Savitzky-Golay filter.





Both descent velocity profiles have a similar shape. At the top of the profile (when the balloon just burst) the payload

accelerates within a few hundred meters to reach its maximum speed of approximately 50 m/s (70 m/s) on June 17 (June 18). While descending, the payload decelerates due to the increasing air resistance, reaching 4–6 m/s in the lower troposphere. The descent velocity of the second flight was continuously higher than that of the first flight, most likely caused by the differences between the used parachutes. Since the AirCore needs a certain amount of time to equilibrate with ambient air, a high descent velocity is expected to impact the altitude retrieval based on a pressure equilibrium assumption to a larger extend than a low

descent velocity. Hence, we expect the resulting vertical profile of the second flight to be stretched more to higher altitudes than the one of the first flight.

**3.2 Altitude attribution vs signal gas release altitude**

The Picarro analyzers mixing ratio time series was attributed to the meteorological flight data according to Sect. 2.1. Figure 5 shows the resulting vertical profile of CO mixing ratios with signal gas spikes on a) June 17 and b) June 18. All eight spikes

that were released at different altitudes (see Table 1) are distinguishable from the baseline data in both flights. Above approximately 20 km the baseline CO is enhanced due to mixing with FG (Engel et al., 2017) and signal gas. Below approximately 12 km the baseline CO is constantly higher than between 12 km and 22 km, indicating higher CO mixing ratios in the troposphere than in the stratosphere. The signal gas spikes are fitted with a Gaussian distribution and the position of the maximum is identified to be the retrieved signal altitude. Table 2 lists the differences between the GPS release altitudes from

the datalogger and the retrieved signal altitudes Δh.

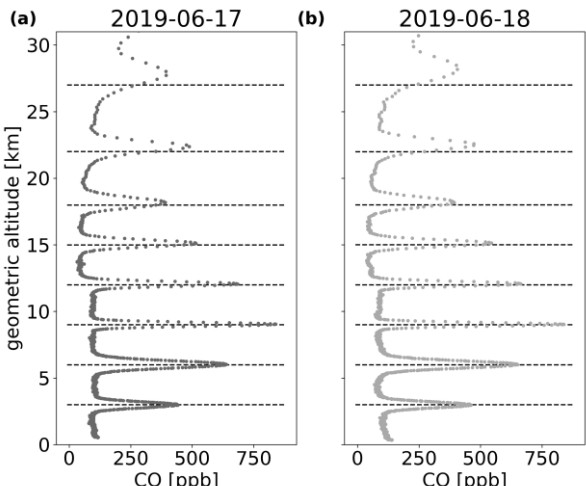

**Figure 5:** CO vertical profiles with signal gas spikes. **(a)** June 17 and **(b)** June 18, Traînou 2019. The dashed lines indicate the signal release GPS altitudes. CO measurements are attributed to geometric retrieval altitudes assuming an instantaneous pressure equilibrium between AirCore and ambient air during sampling.



**Table 2:** Differences between GPS release altitudes and retrieved signal altitudes $\Delta h$.

| Spike number | 1 (27 km) | 2 (22 km) | 3 (18 km) | 4 (15 km) | 5 (12 km) | 6 (9 km) | 7 (6 km) | 8 (3 km) |
|---|---|---|---|---|---|---|---|---|
| $\Delta h$ (June 17) / km | 0.86 | 0.38 | 0.2 | 0.13 | 0.08 | 0.04 | 0.03 | 0 |
| $\Delta h$ (June 18) / km | 1.19 | 0.54 | 0.23 | 0.13 | 0.09 | 0.04 | 0.03 | -0.01 |

For the altitude retrieval, we assume an instantaneous pressure equilibrium. Making this assumption, the sampling altitude is overestimated more at high altitudes than at low altitudes. The overestimation is more pronounced for the flight on June 18 which had higher descent velocities. There are three major effects that lead to this: (i) The small inner diameter of AirCores,

the closing valve and the sample drier in general constitute a flow restriction to the inflowing air. The difference between AirCore pressure and ambient pressure is thus expected to be higher for higher descent velocities. (ii) The descent velocity is especially high at high altitudes for AirCores with a parachute deployed to a weather balloon, as the low ambient pressure leads to a smaller drag by the parachute. (iii) The absolute change in ambient pressure per kilometer is lower at higher altitudes. Hence, even a small difference between AirCore pressure and ambient pressure can lead to a large overestimation at high

altitudes.

As described in Sect. 3.1, the descent velocity during flight 2 on June 18 was generally higher than during flight 1 on June 17. In agreement with the considerations above, $\Delta h$ is larger for spikes above 20 km on June 18 than on June 17. Below 20 km, $\Delta h$ is comparable for both flights with less than 250 m. Between 20 km and 27 km $\Delta h$ is up to approximately 1200 m. Since there were no other relevant differences in the flight parameters, differences in $\Delta h$ between both flights can be attributed to the

differences in descent velocities. We want to emphasize, that the results of these two CO-spiking flights are explicitly tied to the geometry of the GUF AirCores with a fast descent on a parachute, assuming an instantaneous pressure equilibrium and cannot be transferred to other AirCore geometries.

### 3.3 Empirical altitude correction of AirCore profiles

One great potential of the AirCore technology is that it can be deployed to small, cheap and easy to launch weather balloons.

This allows for measurements on a regular basis. Albeit, this involves dealing with high descent velocities especially in the stratosphere with implications for the vertical profile retrieval as shown in Sect. 3.2. It is desirable to have a method for correcting vertical profiles derived from AirCore measurements. The CO-spiking experiment can directly be used to correct the associated vertical profile. However, it is based on injecting small amounts of signal gas at the inlet of the AirCore during sampling, contaminating multiple parts of the atmospheric sample. We tested several parameters, obtained via the CO-spiking

experiment, in order to find a way to correct clean vertical profiles derived from AirCore flights without the CO-spiking set-up. As shown in Sect. 3.2 the absolute difference in altitude between signal altitude and release altitude $\Delta h$ varies between flight 1 and flight 2. Hence, a simple altitude offset correction would neglect the large impact of the descent velocity on $\Delta h$, when assuming an instantaneous pressure equilibrium. The actual pressure inside of the AirCore lags behind the changing ambient





pressure during descent. This time lag $\Delta t$ is observable via the CO-spiking experiment for each spike. It is the flight time of
the AirCore between the theoretically retrieved altitude and the signal release altitude. Figure 6 shows $\Delta t$ as a function of
geometric altitude for both flights. Remarkably, $\Delta t$ does not vary much between the both CO-spiking measurement flights. We
applied a linear fit to the $\Delta t$–altitude relation for each flight. The resulting slope is $0.96 \pm 0.05$ s km$^{-1}$ ($0.91 \pm 0.06$ s km$^{-1}$) for
the flight on June 17 (June 18). Since both slopes are within one standard deviation of the respective other one, we concluded
that differences between both $\Delta t$–altitude relations are insignificant and therefore applied a linear fit to the combined $\Delta t$–
altitude dataset, resulting in a slope of $0.94$ s km$^{-1}$. The maximum difference between the fit and the data is 2.5 s, which is
close to the interval between two flight data records of 1 s and corresponds to an uncertainty of 150 m for a descent velocity
of 60 m/s and of 25 m for a descent velocity of 10 m/s. We used the fitting parameters (slope and intercept) to correct each of
the two vertical profiles, by gradually shifting the sampling time series from Sect. 2.1, step (i): Firstly, for each record i of the
sampling time series, $\Delta t_i$ was calculated as a linear function of the corresponding altitude, using the fit parameters mentioned
above. Secondly, the amount of sample for each record i was updated with the amount of sample, that was calculated via the
ideal gas law for the record $\Delta t_i$ earlier in the time series. Figure 7 shows the resulting corrected vertical profiles of the CO
measurements, with the same fit parameters applied to both profiles. All eight CO spikes in both profiles match the release
altitudes within less than 100 m. We also tested applying the fit parameters obtained from only fitting the respective other
flight's observed $\Delta t$-altitude relation. Again, all eight CO spikes in both profiles match the release altitudes within 120 m.

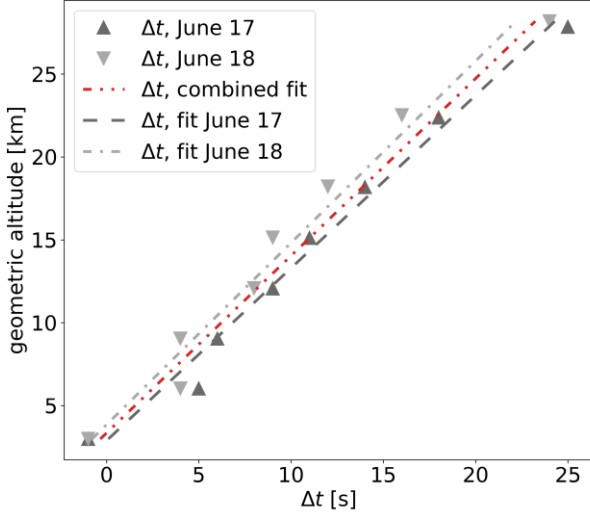


**Figure 6:** Time lag $\Delta t$ between retrieved geometric altitude and signal release GPS altitude.

Despite comprising only two measurement flights, our findings strongly suggest, that $\Delta t$ is a robust empirical parameter which
is characteristic for a specific AirCore and applicable to flights with different descent velocity profiles. The CO-spiking
experiment thus may be used to characterize a specific AirCore geometry, in order to apply an empirical correction to altitude
retrievals based on assuming an instantaneous pressure equilibrium. Once an AirCore is characterized via the CO-spiking





experiment, all vertical profiles from different flights of this AirCore could be empirically corrected, without contaminating the atmospheric air sample with signal gas. Albeit, further measurement flights need to be conducted in order to verify if this hypothesis holds true for flights with other maximum altitudes and other ambient conditions (e.g. temperature profiles) and other AirCore geometries. In particular, this relationship could also change for the same AirCore when a different drier is used

with significantly different flow restriction.

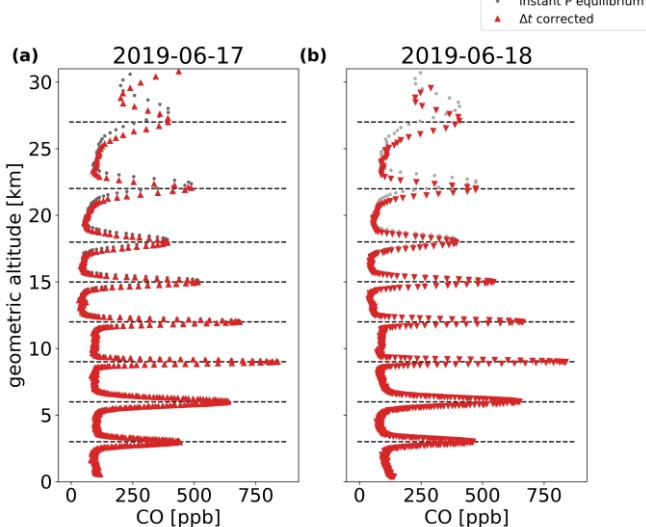

**Figure 7:** Corrected CO vertical profiles with signal gas spikes on **(a)** June 17 (red triangles pointing upwards) and **(b)** June 18 (red triangles pointing downwards), Traînou 2019. The dashed lines indicate the signal release GPS altitudes. CO measurements are attributed to geometric retrieval altitudes assuming an instantaneous pressure equilibrium between AirCore and ambient air during sampling (data from Figure 5

shown as grey circles). The individual profiles were corrected with $\Delta t$ from the combined data.

### 3.4 Modelled vertical resolution vs. signal width

The vertical resolution of a trace gas profile retrieved from an AirCore measurement mainly depends on mixing in the analyzer cell, Taylor dispersion and molecular diffusion inside of the AirCore (Engel et al., 2017; Karion et al., 2010; Membrive et al., 2017). The theoretical altitude resolution has been modelled for $CO_2$ for the GUF AirCore (Engel et al., 2017; Membrive et

al., 2017). Membrive et al. (2017) used a Gaussian kernel with the theoretical altitude resolution of the GUF AirCore for $CO_2$ and $CH_4$ to degrade their measured high resolution AirCore profile and compare it to the actual measured lower resolution GUF AirCore profile. They found a very good agreement between the two $CH_4$ profiles, validating the theory behind AirCores. Albeit, their final results only include profile data down to 200 hPa, corresponding to altitudes well below 13 km.

The CO-spiking system can be used to experimentally quantify the vertical resolution of the CO profile of an AirCore

measurement flight and to directly compare it to the theoretical altitude resolution. The volume of signal gas is of the order of ¼ $ml_n$ per spike and thus considered very small compared to the total sample volume of 1.4 $l_n$, respectively the stratospheric sample of 100 $ml_n$ above 18 km, with respect to the GUF AirCore. The time that the spiking valve is opened is also very short (20–100 ms), so that the original spiking signal can be considered to be of negligible width. Diffusion, Taylor dispersion and





mixing in the analyzer cell broaden the signal to a Gaussian-like shaped signal gas spike. The Gaussian signal gas spike
standard deviation serves as a measure for the altitude resolution for the measurement flights. We used the same approach as
Membrive et al. (2017) and Engel et al. (2017) to calculate the theoretical vertical resolution of the CO profile, taking into
account a 2 h time lag between landing and analysis, the molecular diffusivity of CO in air at 0 °C and 1 atm of 0.18 (Massman,
1998), an effective analyzer cell volume of 6 scc and the in situ ambient pressure and AirCore coil temperature profiles.

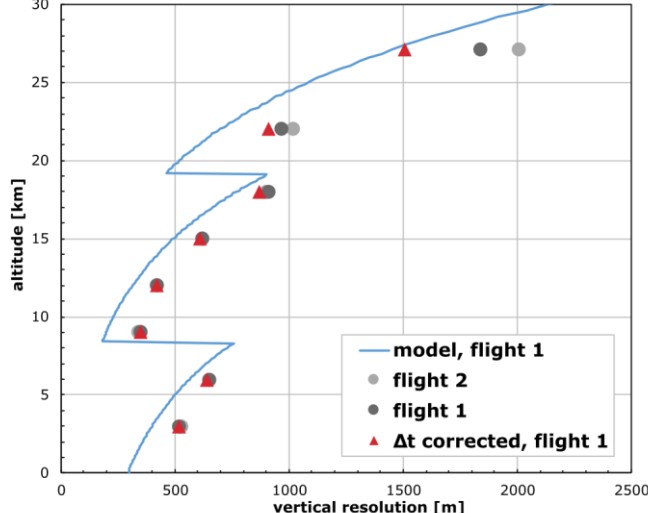

**Figure 8:** Modelled (blue line), uncorrected (flight 1: dark grey circles, flight 2: light grey circles) and corrected (only flight 1: red triangles)
vertical resolution of GUF003.

Figure 8 shows the modelled vertical resolution profile as a function of altitude, the Gaussian standard deviation of the
respective peak and the Gaussian standard deviation derived from the Δt-corrected profile from June 17. Regarding the second
flight on June 18, only data from the uncorrected profile are shown, since the model data and the Δt-corrected data vary only
within 30 m between the two flights. As the Δt correction leads to a compression of the vertical profile, the vertical resolution
of the corrected profiles is better than that of the uncorrected profiles. In general, the vertical resolution is coarser for higher
altitudes, since the amount of sample is lower for higher altitudes. At the top of the profile, the dominating effect is mixing in
the analyzer cell (Engel et al., 2017). Below 19 km (resp. 8 km), the effect of molecular diffusion on the vertical resolution is
larger, since the sample is stored in wider tubing. The modelled vertical resolution is generally less coarse than the vertical
resolution derived from the Δt-corrected profiles, however agrees well within less than 220 m throughout the profile. This
small discrepancy is probably caused by the simplified assumptions guiding the model calculations, that neglect diffusion and
Taylor dispersion during the AirCore sampling process. In addition, the junctions between different diameter parts of the
AirCore might induce additional mixing. Regarding the uncorrected vertical profiles for both measurement flights, this
discrepancy becomes larger at high altitudes, where the peak shape is stretched towards higher altitudes, owing to the
disequilibrium between AirCore and ambient pressure. We only observe this for the two peaks above 20 km, since the
Δt-correction mostly affects the upper parts of the profiles.



## 4. Conclusions

We developed, tested and conducted an altitude dependent CO-spiking experiment, that can be used to quantitatively evaluate different combinations of AirCore geometries and retrieval procedures. It was deployed to a GUF AirCore and used to pulse
small amounts of signal gas in the inlet of the AirCore during descent at predefined GPS altitudes from two weather balloon flights in Traînou in June 2019. The CO trace gas profiles were retrieved assuming an instantaneous pressure equilibrium during the descent of the AirCore and by applying a newly introduced approach to identify an accurate starting point of the AirCore in the CO measurement time series. The comparison of the retrieved signal gas spikes with the actual signal release altitudes show a good agreement throughout the profile with Δh being better than 250 m below 20 km. At higher altitudes the
altitude of the spikes is systematically overestimated in our retrieval. This overestimation reaches up to 900 m (1200 m) at 27 km for the flight on June 17 (June 18). Both flights showed high descent velocities (up to 50 m/s, resp. 70 m/s) especially in the stratosphere, that differed strongly between both flights, therefore representing very different sampling conditions. The actual pressure inside of the AirCore lags behind the changing ambient pressure during descent. In case of our AirCore, we identified this time lag Δt to be a possible empirical correction parameter, that increases linearly with altitude and seems to be
independent of the descent velocity and therefore stable among different flights. The corrected profiles showed an excellent agreement with the actual release altitudes within 120 m, even if the correction parameters derived from the respective other flight were applied. Further measurement flights need to be conducted with the CO-spiking system in order to test for the scope of validity of Δt as a robust empirical correction parameter, regarding different ambient conditions and maximum flight altitudes. Again, we emphasize that this correction will be specific for each AirCore, or at least AirCore geometry. Albeit, our
findings strongly suggest, that an AirCore geometry and altitude dependent empirical Δt-correction may be applied to AirCore profiles even if the payload was without the CO-spiking system, once the relation has been established for a particular set-up. This implies the possibility to derive trace gas profiles from AirCore measurement flights with an optimally improved altitude attribution even at high altitudes above 20 km without the need for inclusion of a spiking system during each flight. This still allows for easier operation and also provides continuous vertical profiles that have not been affected by signal gas injection.
We calculated the theoretical vertical resolution for both flights from in situ parameters including the AirCore coil temperature following Membrive et al. (2017) and compared it to the Gaussian standard deviation of the signal gas spikes. This Gaussian standard deviation serves as a measure for the in situ vertical resolution of the AirCore profile. The modelled vertical resolution is too optimistic compared to the vertical resolution derived from the Δt-corrected profiles, however agrees well within less than 220 m throughout the profile. This discrepancy is probably caused by the simplified assumptions guiding the model
calculations. Albeit, the magnitude of the experimentally derived vertical resolution and the general shape of the resolution–altitude relation can be reproduced by the model.

Our results based on the newly developed CO-spiking system proof, that trace gas profiles can be obtained from AirCores deployed to low-cost weather balloons with a highly accurate altitude attribution at least up to 27 km and a fine vertical resolution, which is close to the calculations of a simple model. The quantities for Δh and the vertical resolution derived from

our measurement flights are strictly bound to the GUF AirCore geometry in combination with the pressure equilibrium assumption guiding the data processing, respectively the applied Δt-correction. As an alternative to assuming an instantaneous pressure equilibrium, an altitude attribution approach has been suggested (P. Tans, NOAA, private communication, 2020), that is based on modelling the pressure drop across the AirCore during sampling and the flow of air into the AirCore. When such a retrieval procedure is established, one could check if Δt is a valid correction parameter and needed for profiles retrieved this

way. The CO-spiking technique can be deployed to any AirCore and used to compare and evaluate different altitude retrieval procedures in combination with different AirCore geometries and flight platforms in future studies.

**Code availability**

The Python Software Code for data processing and evaluation can be made available by the corresponding author upon request. The C++ Code that is compiled to the Arduino MEGA 2560 micro controller can be made available by the corresponding

author upon request.

**Data availability**

Data are available from the corresponding author upon request in ICARTT format.

**Author contribution**

TW, AE and RS were involved in developing the CO-spiking system set-up and performing the field measurements. TW and

AE were involved in the data evaluation and interpretation. TW wrote the article and improved the retrieval software in collaboration with AE. AE designed this study.

**Competing interests**

The authors declare that they have no conflict of interest.

**Acknowledgements**

The latest work of the University of Frankfurt on AirCore has been funded through the EU Infrastructure Project RINGO (Grand agreement no. 730944). We would like to thank the team from LSCE in particular Thomas Laemmel for hosting the AirCore campaign in Traînou in 2019 and Huilin Chen for coordinating the RINGO AirCore activities. Many thanks also to Colm Sweeney and Bianca Baier for helpful discussions on Picarro measurements. We would like to thank Rainer Rossberg and Audrey Goujon for the cooperation on developing the CO-spiking control electronics. We further thank Irina Kistner for



her help in preparing and performing the AirCore CO-spiking flights. The support of the workshops and technicians at the University of Frankfurt is gratefully acknowledged. We would like to thank the Fritz Gyger AG in particular Erhard Würsten for technical support regarding the micro valve.

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
