# Peer review of "Testing the altitude attribution and vertical resolution of AirCore measurements with a new spiking method"

_Atmospheric Measurement Techniques, 2020_

## Author Comment (AC1)

**Reply to RC1**

Manuscript information:
- Title: Testing the altitude attribution and vertical resolution of AirCore measurements with a new spiking method
- Author(s): Thomas Wagenhäuser, Andreas Engel, Robert Sitals
- MS No.: amt-2020-461
- MS type: Research article
- Iteration: Final response (AMT Discussions)

We would like to thank Pieter Tans for the constructive comments. In the following document, the reviewers' comments are marked in *italic font* and indented, our answers are in regular font. Changes in the manuscript are marked-up in red and listed as framed screenshots below the respective comment. For clearness: The line numbers in the RC refer to amt-2020-461-manuscript-version2.pdf instead of the preprint version. The two version only differ by minor formatting aspects. The line numbers in our listed changes refer to the marked-up version of the revised manuscript, that is provided separately.

**Point-by-Point reply**

1. *Line 81: The internal diameter is what matters, so please provide that. I assume that the predicted results use the ID.*

Agreed. The internal diameters are used to model the vertical resolution.

> 85    Three thin-walled stainless steel tubes with different  diameters (internal diameters: 1.76 mm, 3.6 mm, 7.6 mm;  outer diameters: 2 mm, 4 mm, 8 mm), coated with silconert2000® and soldered together result in the 100 m long and coiled GUF AirCore. This design allows to rapidly collect air in the large-diameter tubing, which is then gradually pushed into the

2. *line 137: The fill gas at the closed end of the tube will not be distributed as a Gaussian. It has to be asymmetric because the end is closed off. What it looks like depends on how much fill gas is left, but close to the end the spatial derivative of each gas fraction has to go to zero. If there is a lot of fill gas left, occupying several diffusion length scales, the fill gas fraction must approach 1 at the closed end. When the fill gas enters the analyzer the transition ought to be rapid, unless the tubing toward the analyzer and the analyzer cell itself cause a lot of mixing. Some years ago I analyzed experiments with "plug" transitions, sudden mole fraction shifts inserted very close to the Picarro. In the hypothetical case that the cell would be perfectly mixed all the time, the insertion of a plug should produce a negative exponential approach toward the new steady state. If there is plug flow within the cell, so that the rapid transition is mostly preserved, the approach to the new state should be linear. It turns out that the actual transition was in between these two cases. The "response function" of each analyzer will depend on pressure, cell volume, and shape of the cell. This subject comes back on line 156.*

Thank you for your thoughts on this. We agree, that the fill gas will not be distributed as a Gaussian, which however had been assumed in the past by Engel et al. 2017. That is why we decided to introduce the new approach to identify an accurate starting point by reconstructing the gas fraction time series. In order to make this clearer, we inserted an additional citation in the manuscript:

in the AirCore (high CO), which is fourthly followed by the stratospheric sample (STRAT, low CO). The resulting idealized
160  CO mixing ratio time series including gradual transitions between gas fractions is shown in Figure 2a. In the past, a Gaussian
distribution was fitted to the FG peak in the CO measurements (Engel et al., 2017). The half maxima of the fit were then
considered the start of the AirCore, respectively the start of the STRAT, as described in Engel et al. (2017) Sect. 2.4.1 and
2.4.3. This FG peak however is partly mixed with PG, Cal gas and STRAT. In the new approach, we reconstruct the gas

In the cases that we tested with our measurement setup, the fitting algorithm with CDFs of a Gumbel distributions was able to reproduce the asymmetric shape of the fill gas peak and yielded satisfying results for each gas fraction transition – starting from very close to 0 (but not exact 0) and approaching 1 with increasing time (see Fig. 2c). We addressed this issue in the revised version of the manuscript:

where μ is the mode of the respective Gumbel distribution (i.e. the inflection point of the CDF) and β is a measure for the
standard deviation. Strictly spoken, the CDF of a Gumbel distribution does not start from exactly 0 – which in reality should
190  be the case for e.g. the FG fraction due to being closed off. Albeit, we found that it reliably generates satisfying results. For
simplicity of the fitting process we decided to use the CDF of the Gumbel distribution instead of the CDF of the lognormal
distribution. By altering the parameters $\mu_i$ and $\beta_i$ simultaneously for each gas taking into account Eq. (85), a CO mixing ratio

The maximum of the fill gas fraction is indeed reduced below a value of 1 occasionally with our measurement setup. The idealized time series shown in Figure 2 was actually generated using data from an GUF AirCore measurement in Sodankylä in June 2018. Figure 2 reveals, that the measured mole fraction at the maximum of the fill gas peak is the result of mixing with both, Cal gas (due to Taylor dispersion in the transfer line and mixing in the analyzer cell) and stratospheric sample (due to Taylor dispersion, diffusion and mixing in the cell).

3. *line 193:  I suppose the signal gas mixture is CO-in-natural air. If a larger spike is inserted, you don't want to alter the main gases of interest.*

Thank you for your input. The amount of $CO_2$ and $CH_4$ in the signal gas is not relevant for the evaluation of the altitude retrieval, which is the aim of this paper. We do not recommend using contaminated sections of the AirCore profile for atmospheric interpretation at all. That is why we decided not to mention the $CO_2$ and $CH_4$ content of the signal gas. If we were to permanently include the CO-spiking system in an AirCore setup, we could however reduce the number of CO-spikes e.g. to just one (I would then suggest at the top of the profile) in order to minimize the effect of contamination due to mixing with signal gas. We included a short statement and mentioned the new idea at the end of sect. 3.3 in the revised version of the manuscript:

other AirCore geometries. In particular, this relationship could also change for the same AirCore when a different drier is used
340  with significantly different flow restriction. We do not recommend using contaminated sections of the AirCore profile for
atmospheric interpretation. Nevertheless, the CO-spiking system could also permanently be included in a once fully
characterized AirCore setup and used to insert only one spike at the top of the profile, in order to obtain highly accurate trace
gas profiles (including other trace gases than CO measured with the Picarro analyzer) with minimal signal gas contamination.

4. *line 234:  typo - should be June 17*

Yes, changed that. Thank you.

5. *line 245: "resulting calculated vertical profile"  (this is for clarity)*

Done.

6. *line 350:  I would be very surprised if junctions between sections could cause much additional mixing. The flow conditions in the tube are extremely far away from any kind of turbulence.  My suggestion is to look further into the analyzer contribution to mixing.*

Thanks for pointing that out. We took this into account in the revised version of the manuscript:

profiles, however agrees well within less than 220 m throughout the profile. This small discrepancy is probably caused by the simplified assumptions guiding the model calculations, that neglect diffusion and Taylor dispersion during the AirCore sampling process. In addition, the effect of mixing in the analyzer cell may be underestimated in the model and the junctions between different diameter parts of the AirCore might induce a small amount of additional mixing. Regarding the uncorrected

385 vertical profiles for both measurement flights, this discrepancy becomes larger at high altitudes, where the peak shape is

7. *line 387:  I suggest replacing "proof," with "prove"*

Done.

---

## Author Comment (AC2)

**Reply to RC2**

Manuscript information:
- Title: Testing the altitude attribution and vertical resolution of AirCore measurements with a new spiking method
- Author(s): Thomas Wagenhäuser, Andreas Engel, Robert Sitals
- MS No.: amt-2020-461
- MS type: Research article
- Iteration: Final response (AMT Discussions)

We would like to thank William Sturges for the constructive comments. In the following document, the reviewers' comments are marked in *italic font* and indented, our answers are in regular font. Changes in the manuscript are marked-up in red and listed as framed screenshots below the respective comment. The line numbers in our listed changes refer to the marked-up version of the revised manuscript, that is provided separately.

**Point-by-Point reply**

1. *My only comment would be that, for the sake of anyone **not** working directly on AirCores, this would benefit from having a little more explanatory text. E.g. under Section 2.1, a clearer summary of these steps that does not require reference to Engel et al. would make reading/understanding much easier.*

Thank you for your constructive comment. We added more explanatory text to Section 2.1 in the revised version of the manuscript:
* * *
ordered and refined compared to the four-stage process described in Engel et al. (2017). The overall concept of the retrieval remains the same. (i) The sampling of air during the balloon descent is calculated based on the ideal gas law.
* * *
120  (ii) The start and end times of the AirCore measurement in the analyzer time series are determined. (iii) The sampling and the analysis can be matched based on the molar amount . Steps (i) and (iii) are still performed according to Engel et al. (2017) and are shortly described in the following.

(i): Under the assumption of an instantaneous pressure equilibrium the molar amount $n$ of an ideal gas within a constant AirCore volume $V$ at the sampling time $t$ can be described by the ideal gas law:

$$n(t) = \frac{p(t) \cdot V}{R \cdot T(t)}, \tag{1}$$

125 where $p$ and $T$ are the atmospheric pressure, resp. the mean AirCore temperature at $t$ and $R$ is the general gas constant. The relative amount of gas $n_{rel}$ is then described by:

$$n_{rel}(t) = \frac{n(t)}{n(t_{close})} = \frac{p(t) \cdot T(t_{close})}{p(t_{close}) \cdot T(t)}, \tag{2}$$

with the total molar amount of gas $n(t_{close})$ at the closing time $t_{close}$ of the AirCore. The data evaluation software takes into account special cases, where air can be lost during sampling.

130 (iii): Since the analyzer is operated at a constant mass flow, the relative amount of measured gas $m_{rel}(t')$ at the elapsed measurement time $t'$ can be described as

$$m_{rel}(t') = \frac{t'}{t'_{end}}, \tag{3}$$

with the total AirCore measurement time $t'_{end}$. By interpolating the meteorological and positional data collected during sampling from $n_{rel}(t)$ to $m_{rel}(t')$, it is attributed to the trace gas measurements $\chi(t')$.

135 In order to accurately derive $m_{rel}(t')$, the start and end points of the AirCore analysis in the continuous Picarro measurement time series need to be known (step (ii)). In Sect. 2.2 we present a new approach to determine the start point of the AirCore in the measured trace gas time series. This new approach has the advantage of providing an objective start point without the need for subjective judging.
* * *
We also updated the subsequent numeration of equations (not shown here).

Done.

*3. L.16 "shown" not "uncovered"*

Done.

*4. L.17 "to be represented by possible empirical"*

Thanks for your suggestion. We decided to put the statement in different words to make it more accurate in the revised version of the manuscript:

> 15 within 250 m below 20 km. Above 20 km the positive bias becomes larger and reaches up to 1.2 km at 27 km altitude. Differences in descent velocities are  shown to have a major impact on the altitude attribution bias.  We parameterize the time lag between the theoretically attributed altitude and the actual CO-spike release altitude for both flights together and use it to empirically correct
>
> 20 our AirCore altitude retrieval. Regarding the corrected profiles, the altitude attribution is accurate within $\pm 120$ m throughout

*5. L.19 is it +/- 120 m or +120 offset?*

It's ±120 m. Added this.

*6. L.50 "needs to be attributed to positional data" – doesn't it just need altitudinal data? Lat/Long you'd get from GPS, wouldn't you?*

Thanks for your suggestion. Indeed, the statement in the original manuscript was inaccurate. We decided to stick to the term "positional", since this statement is not restricted to passive AirCore sampling but also holds true for active AirCores. In all cases it's the molar amount of gas during the sampling process that is matched to the analysis time series. GPS altitude is not needed for this process, albeit it is one of the desired variables. It is not treated differently from Lat/Long data in this matching process. We clarified this in the updated version of the manuscript:

> 50 in 2015 and to small weather balloons flown in 2016.
> The wide range of platforms and fields of application concerning AirCore sampling (regardless of being active or passive) all have one in common: a continuous sample of atmospheric air is collected together with meteorological and positional data, that need to be attributed to the trace gas measurements of the sample after analysis.  Regarding vertical profiles from passive AirCores, an altitude attribution

*7. L.90 what is the push gas made of?*

We provided relevant information in the updated version of the manuscript:

> 90 comprises up to 8 temperature sensors, a pressure sensor, a GPS-antenna, an SD-card holder for data logging and controls the closing valve.
> Before launch, the AirCore is flushed with fill gas (FG, air standard with known trace gas mole fractions) and sealed at one end. During ascent it empties due to the decreasing ambient pressure with height. A small amount of FG remains in the AirCore.

And three lines below:

> push gas (PG) and analyzed with a Picarro G2401 CRDS continuous gas analyzer for $H_2O$, $CO$, $CH_4$ and $CO_2$ mole fractions at a constant rate. FG and PG are identical in the GUF setup. Figure 1 shows the analytical set-up for the measurement process.

*8. Fig. 1 is quite tough to follow unless you have a little more background*

Thanks for pointing that out. We added explanatory text in the updated version of the manuscript:

> push gas (PG) and analyzed with a Picarro G2401 CRDS continuous gas analyzer for $H_2O$, CO, $CH_4$ and $CO_2$ mole fractions at a constant rate. FG and PG are identical in the GUF setup. Figure 1 shows the analytical set-up for the measurement process. In the bypass/flushing position of the two position valve (Fig. 1a), PG is measured and the open transfer lines to the AirCore are flushed. After connecting the AirCore to the transfer lines, the two position valve is switched to measurement position
> 100 (Fig. 1b), so that the sample is pushed through the analyzer. Since June 17, 2019, our Picarro analyzer operated in the inlet

*9. L.102 what does "PG resp. a calibration standard" mean? I didn't understand this.*

We added labels to the transfer lines in Fig. 1a and referred to them in the caption in the updated version of the manuscript in order to make it more comprehensible:

[Figure]

> **Figure 1:** Analytical set-up for AirCore measurements. Pressure is controlled by the digital pressure controller (DPC). Compared to the previously published set-up by Engel et al. (2017) the mass flow controller has been replaced by a needle valve (NV). The Picarro operates in inlet control mode. In the bypass/flushing position **(a)** push gas (PG) is measured bypassing the AirCore while the transfer lines (TL) are
> 110 being flushed: $TL_P$ with PG and $TL_C$  with a calibration standard (Cal gas). Tubing that contains PG is indicated in blue, Cal gas in orange. In the AirCore measurement position **(b)** the PG is passed through the AirCore and pushes the air to the Picarro. Directly after

*10. L104 I am not clear about "only tubing involved at the start of the AirCore measurement is coloured"; what is meant by "involved" – it's all involved isn't it?*

Thanks for pointing that out. We rephrased it in the updated version:

> orange. In the AirCore measurement position **(b)** the PG is passed through the AirCore and pushes the air to the Picarro. Directly after switching to **(b)** a small amount of Cal gas is measured that has been enclosed by the transfer line $TL_C$. For clearness, in **(b)** only tubing  containing gas that is measured at the  beginning of the AirCore  analysis is coloured (adapted from Engel et al., 2017).

*11. L.117 not clear what "starting point in the analysis" refers to.*

Thanks for pointing that out. We rephrased it in the updated version, in order to make it clearer:

> 140 Membrive et al. (2017) stated that for their slowly descending high resolution AirCore the dominating uncertainty source in the stratosphere is related to  selecting the start point of the AirCore analysis  in the continuous Picarro measurement time series. They link this to the low amount of stratospheric sample compared to the tropospheric sample. For AirCores with less stratospheric sample the effect can be considered to be larger. Until now, the choice of the start point relied on subjective judging (Engel et al., 2017; Membrive et al., 2017). In order to systematically
> 145 evaluate the altitude attribution procedure with the CO-spiking experiment presented in this paper, as many as possible subjective parameters need to be eliminated. We therefore decided to refine the process of selecting the start time of the AirCore and introduce a new approach to identify an accurate start point without the need for subjective judging.

**12. L.129 how high is high CO?**

Good point. We added information about standard gas CO mixing ratios from recent campaigns for clarity and improved our description of the measurements in the updated version of the manuscript:

> For a regular GUF set-up AirCore flight analysis we first measure PG (high CO, 1.4 ppm in the recent GUF campaigns). We
> then switch the two position valve (see Figure 1b) like described in Engel et al. (2017) so that secondly the measurements
> 155 gradually approach the low CO mixing ratio of the calibration gas (Cal gas, 0.2 ppm in the recent GUF campaigns) that was
> left in the transfer line $TL_C$ between AirCore and analyzer. the calibration gas (Cal gas) within the transfer line $TL_C$ between

**13. L.131 maybe explain how "Cal gas is used to distinguish between PG and FG"?**

We extended the statement in the updated version of the manuscript:

> AirCore and analyzer is measured (low CO). Since in our setup one standard gas is used as both PG and FG, Thethe Cal gas
> is usedserves to distinguish between PG and FG in the measurement time series. Thirdly, it is followed by the remaining FG
> in the AirCore (high CO), which is fourthly followed by the stratospheric sample (STRAT, low CO). The resulting idealized
> 160 CO mixing ratio time series including gradual transitions between gas fractions is shown in Figure 2a. In the past, a Gaussian

**14. L.189 what does "fastening valve" mean? I've not heard of this before (shutoff valve?).**

Instead of "fastening type" we now call it "mounting hardware":

> **Figure 3:** Fastening typeMounting hardware for the SMLD 300G micro valve. (1) micro valve, (2) valve coil, (3) O-ring (material: viton), (4) inlet adapter, (5) valve holder. Fritz Gyger AG 2020.
>
> 215 The CO-spiking set-up consists of a signal gas reservoir, a micro valve and a connector which directly connects the micro
> valve to the open end of the AirCore in front of the sample drier. The adaptor is designed to have a negligible flow resistance
> for sampled air, while inducing only a minimal dead volume to the sampling system. We used the micro valve SMLD 300G
> by Gyger, which is light-weight and suited to dose signal gas volumes of around ¼ $cm^3_n$ on the time scale of 20–
> 100 milliseconds, thereby influencing the sampling process during descent as little as possible. Figure 3 illustrates the fastening
> 220 typemounting hardware for the micro valve.

**15. Fig. 8 It took a while for me to realise that the steps in the curve related to the three diameters of tubing - maybe point this out from the start?**

Thank you for your feedback. This has also been pointed out by Anna Karion. We added one sentence for clarification:

> **Figure 8:** Modelled (blue line), uncorrected (flight 1: dark grey circles, flight 2: light grey circles) and corrected (only flight 1: red triangles)
> 370 vertical resolution of GUF003.
>
> Taking into account the three different inner diameters and lengths of GUF AirCore tubing, the modelled vertical resolution
> exhibits two steps corresponding to the junctions between two adjacent parts of tubing. Figure 8 shows the modelled vertical
> resolution profile as a function of altitude, the Gaussian standard deviation of the respective peak and the Gaussian standard
> deviation derived from the Δt-corrected profile from June 17. Regarding the second flight on June 18, only data from the

---

## Author Comment (AC3)

**Reply to RC3**

Manuscript information:
- Title: Testing the altitude attribution and vertical resolution of AirCore measurements with a new spiking method
- Author(s): Thomas Wagenhäuser, Andreas Engel, Robert Sitals
- MS No.: amt-2020-461
- MS type: Research article
- Iteration: Final response (AMT Discussions)

We would like to thank Anna Karion for the constructive comments. In the following document, the reviewers' comments are marked in *italic font* and indented, our answers are in regular font. Changes in the manuscript are marked-up in red and listed as framed screenshots below the respective comment. The line numbers in our listed changes refer to the marked-up version of the revised manuscript, that is provided separately.

**Point-by-Point reply**

1. *L188, perhaps I missed this earlier but what is ml_n ? (and again elsehwere including L326, after 1.4 liters (ln?))?*

Thanks for pointing that out. We added an explanation and changed volumetric units to SI units in the updated version of the manuscript:

> 100 (Fig. 1b), so that the sample is pushed through the analyzer. Since June 17, 2019, our Picarro analyzer operated in the inlet valve control mode at a constant rate at ~30 $cm^3_n$ $min^{-1}$ (n indicates normal conditions: 1013 hPa, 0 °C)seem for AirCore measurements. This is similar to previously published operating methods (e.g. Andersen et al., 2018; Membrive et al., 2017).

> for sampled air, while inducing only a minimal dead volume to the sampling system. We used the micro valve SMLD 300G by Gyger, which is light-weight and suited to dose signal gas volumes of around ¼ $cm^3_n$ on the time scale of 20–100 milliseconds, thereby influencing the sampling process during descent as little as possible. Figure 3 illustrates the mounting hardware for the micro valve.
> 220
> We utilized a modified nylon compression ring and an additional O-ring (Figure 3, (3)) to achieve leak-tightness at low-temperature. In addition, the micro valve is heated during flight in order to remain leak-tight and maintain its functionality. The signal gas reservoir consists of a 2 m tubing (total volume approximately 50 $cm^3$) and a particle filter. The particle filter

> measurement flight and to directly compare it to the theoretical altitude resolution. The volume of signal gas is of the order of ¼ $cm^3_n$ per spike and thus considered very small compared to the total sample volume of 1.00 $cm^3_n$, respectively the
> 360 stratospheric sample of 100 $cm^3_n$ above 18 km, with respect to the GUF AirCore. The time that the spiking valve is opened

2. *L197, rather than "bar" perhaps SI units would be used here (editors can comment on journal policy) (same comment, line 332 using "atm".)*

Changed that from 4 bar to 0.4 MPa and from 1 atm to 1013 hPa.

> 225 packed together with the AirCore in the Styrofoam box and directly connected to the micro valve. It can be pressurized via a valve at the other end of the tube and flushed by activating the micro valve. For flight preparation, the signal gas reservoir is pressurized from a signal gas canister with approximately 0.4 MPa, flushed by activating the micro valve and then pressurized again. The signal gas has a very high mixing ratio of CO (in our case approx. 90 ppm) compared to typical

> gas spike standard deviation serves as a measure for the altitude resolution for the measurement flights. We used the same approach as Membrive et al. (2017) and Engel et al. (2017) to calculate the theoretical vertical resolution of the CO profile,
> 365 taking into account a 2 h time lag between landing and analysis, the molecular diffusivity of CO in air at 0 °C and 1013 hPa atm

3. *L213, approximately should be spelled out here and elsewhere I believe (editors can comment on that)*

Done.

4. *L 229 Typo, June 18 is used twice, should be June 17*

Done.

5. *L243 should be the analyzer's (apostrophe added)*

Done.

6. *Fig 4 and Fig 5, one is labeled GPS Altitude and one geometric - are these the same thing? (i.e. both based on the GPS reading?). (and same question for other figures - perhaps they should all be made with consistent labeling).*

This is an interesting point, that has also been risen by the editor of this manuscript, Fred Stroh (editor review prior to interactive discussion). There is a small difference: In Fig 4 the GPS altitude information is directly linked to the descent velocity (which is calculated from the GPS altitude time series). In Fig 5 (and the following) the GPS altitude data has been attributed to the trace gas mixing ratios that were measured after the flight. In context of this paper it seems appropriate to be specific especially in the latter case (thanks again to Fred Stroh for pointing this out).

We updated the Fig 4 caption and the yaxis label of Fig 8 in order to make this clearer and consistent:

**Figure 4:** Descent velocities for the CO-spiking flights calculated from the GPS altitude time series on June 17 (dark, dashed) and June 18 (light, dash dotted), Traînou 2019. The data is smoothed with a Savitzky-Golay filter.

Both descent velocity profiles have a similar shape. At the top of the profile (when the balloon just burst) the payload

265 accelerates within a few hundred meters to reach its maximum speed of approximately 50 m/ s$^{-1}$ (70 m/ s$^{-1}$) on June 17

[Figure]

**Figure 8:** Modelled (blue line), uncorrected (flight 1: dark grey circles, flight 2: light grey circles) and corrected (only flight 1: red triangles)
370 vertical resolution of GUF003.

7. *L286: "between the both" should be "between the two"*

Done.

8. *L335 and on: It would help the reader if discussion of Figure 8 could mention the sharp changes in the modeled resolution occur at the junctions between different diameter parts of the Aircore.*

Thanks for pointing that out. William Sturges also raised this point. We added one sentence for clarification:

> 370   **Figure 8:** Modelled (blue line), uncorrected (flight 1: dark grey circles, flight 2: light grey circles) and corrected (only flight 1: red triangles) vertical resolution of GUF003.
>
> Taking into account the three different inner diameters and lengths of GUF AirCore tubing, the modelled vertical resolution exhibits two steps corresponding to the junctions between two adjacent parts of tubing. Figure 8 shows the modelled vertical resolution profile as a function of altitude, the Gaussian standard deviation of the respective peak and the Gaussian standard deviation derived from the $\Delta t$-corrected profile from June 17. Regarding the second flight on June 18, only data from the